# Preparation and Recognition Properties of Molecularly Imprinted Nanofiber Membrane of Chrysin

**DOI:** 10.3390/polym14122398

**Published:** 2022-06-14

**Authors:** Yaohui Wang, Long Li, Gege Cheng, Lanfu Li, Xiuyu Liu, Qin Huang

**Affiliations:** 1School of Chemistry and Chemical Engineering, Guangxi Minzu University, Nanning 530006, China; wwangyh970218@163.com (Y.W.); lilong19980227@163.com (L.L.); ggcheng2022@163.com (G.C.); a17876072393@163.com (L.L.); 2Key Laboratory of Chemistry and Engineering of Forest Products, State Ethnic Affairs Commission, Nanning 530006, China; 3Guangxi Key Laboratory of Chemistry and Engineering of Forest Products, Nanning 530006, China; 4Guangxi Forest Product Chemistry and Engineering Collaborative Innovation Center, Nanning 530006, China

**Keywords:** chrysin, electrospinning, molecular imprinting membrane, adsorption

## Abstract

The separation and extraction of chrysin from active ingredients of natural products are of great significance, but the existing separation and extraction methods have certain drawbacks. Here, chrysin molecularly imprinted nanofiber membranes (MINMs) were prepared by means of electrospinning using chrysin as a template and polyvinyl alcohol and natural renewable resource rosin ester as membrane materials, which were used for the separation of active components in the natural product. The MINM was examined using Fourier transform infrared (FT-IR) spectroscopy, scanning electron microscopy (SEM), and thermogravimetric analysis (TGA). The adsorption performance, adsorption kinetics, adsorption selectivity, and reusability of the MINM were investigated in static adsorption experiments. The analysis results show that the MINM was successfully prepared with good morphology and thermal stability. The MINM has a good adsorption capacity for chrysin, showing fast adsorption kinetics, and the maximum adsorption capacity was 127.5 mg·g^−1^, conforming to the Langmuir isotherm model and pseudo-second-order kinetic model. In addition, the MINM exhibited good selectivity and excellent reusability. Therefore, the MINM proposed in this paper is a promising material for the adsorption and separation of chrysin.

## 1. Introduction

In recent years, with the development of analytical methods (high-performance liquid chromatography, ultra-high-performance liquid chromatography, electrochromatography, etc.) and extraction techniques (membrane separation, semi-bionic extraction, high-speed countercurrent chromatography, etc.), pharmacologically active natural products have gained unprecedented popularity [1,2]. Most of them have had profound effects on our lives. Chrysin is chemically known as 5,7-dihydroxy flavone. It is a natural flavonoid and is the main bioactive component isolated from traditional *Oroxylum indicum* [3,4,5]. Chrysin exhibits anti-oxidative [6], anti-viral, immunomodulatory, and anti-inflammatory effects [7,8]. Numerous studies have indicated that chrysin inhibits tumor cell proliferation and induces tumor cell apoptosis, restrains tumor angiogenesis, and reverses tumor cell multi-drug resistance [9,10,11,12]. It is a natural active ingredient with an anti-tumor effect. Therefore, the extraction and utilization of chrysin are of great economic importance.

According to several reviews of the literature, methods such as high-performance liquid chromatography (HPLC) [13], column chromatography [14], chromatography [15,16], adsorption [17,18,19], water-methanol [20,21], and ultrasonic/microwave-assisted extraction [22,23,24] have been developed for the analysis and separation of chrysin in *Oroxylum indicum*. However, most methods, such as column chromatography and other traditional methods, have a low separation effect on the structural analogs of chrysin. After extraction, further separation and purification are required. An efficient, low-cost material for chrysin extraction and purification is currently lacking.

In recent years, molecular imprinting technology [25,26,27] has been generally used for separating and purifying the effective constituents from various natural products. Compared to other methods, the molecular imprinting method has the advantages of stronger affinity and recognition ability. However, molecularly imprinted polymers are usually prepared as a whole material, resulting in most imprinted cavities lying deep within the polymer matrix. Due to these factors, it will have the disadvantages of poor imprinted loci accessibility, incomplete removal of the template, and weaker binding capacity [28,29,30].

Molecularly imprinted film is an effective material used to control templates located at the surface of imprinted materials; a typical example of this surface imprinting, which is carried out by immobilizing template molecules at the surface of suitable substrates, forming thin imprinted films [31,32]. Researchers have used grafting, coating, electrostatic deposition, electrostatic spinning, and other methods to prepare molecularly imprinted membranes (MIMs) to increase membrane flux [33,34,35,36]. Electrospinning nanofiber membranes have the characteristics of a large specific surface area, high porosity, and easy modification [37,38,39]. They have been widely used in tissue engineering, drug delivery, catalysis, wound dressings, and other fields [40,41,42,43,44,45]. Sueyoshi et al. [46] used optically active glutamic acid (Zd-Glu and Zl-Glu) as a template molecule and cellulose acetate as the base membrane to prepare molecularly imprinted membranes by means of electrospinning. The results show that the imprinted membranes prepared by electrospinning had higher permeability and flux than molecularly imprinted membranes prepared by other methods.

In this article, a molecularly imprinted material with excellent recognition and selective absorption for chrysin was prepared. We used polyvinyl alcohol (PVA) fiber as a supporting material and rosin ester as an auxiliary material to fabricate a molecularly imprinted nanofiber membrane (MINM). A detailed examination of MINM adsorption and selective recognition was conducted through the analysis of their kinetics, their isotherms, and their selective adsorption performances.

## 2. Experimental Section

### 2.1. Materials

Chrysin (480-40-0) was purchased from Shanghai Aladdin Biochemical Technology Co., Ltd. (Shanghai, China). Chloramphenicol (56-75-7) and oxytetracycline (79-57-2) were obtained from Shanghai Maclin Biochemical Technology Co., Ltd. (Shanghai, China). Polyvinyl alcohol (PVA, 9002-89-5), N,N-dimethylformamide (DMF, 68-12-2), and acetic acid were obtained from Shanghai Maclin Biochemical Technology Co., Ltd. (Shanghai, China). Methyl alcohol (67-56-1) was obtained from Chengdu Cologne Chemicals Co., Ltd. (Chengdu, China). Methacrylic acid (MAA, 79-41-4) and azobisisobutyronitrile (AIBN, 78-67-1) were obtained from Sinopharm Chemical Reagent Co., Ltd. (Shanghai, China). Ethylene glycol dimethacrylate (EGDMA, 97-90-5) was purchased from Alfa Aesar (Qingdao, China). Ethylene glycol maleic rosinate acrylate (EGMRA) was provided by Wuzhou Sun Shine Forestry & Chemicals Co., Ltd. (Guangxi, Wuzhou, China).

### 2.2. Preparation of Molecularly Imprinted Membranes

#### 2.2.1. Preparation of Molecularly Imprinted Composite Membrane

The molecularly imprinted composite membrane (MICM) was prepared by means of electrospinning after mixing the molecularly imprinted polymer (MIP) and membrane material. The precipitation polymerization method was used for the preparation of the MIP microspheres. Accurately weighing chrysin (0.0675 g) with an electronic balance (Practum124-1cn, Sartorius, Göttingen, Germany), in a 250 mL three-necked flask, chrysin was dissolved in 100 mL of methanol. MAA (0.1825 g), EGDMA (1.6848 g), EGMRA (0.4200 g), and AIBN (0.0453 g) were dissolved in the solution and used an ultrasonic cleaner (KQ-800E, Kun Shan Ultrasonic Instruments Co., Ltd., Kunshan, China) to sonicate the raw materials to fully dissolve them. Hence, nitrogen was immediately added to the mixture, and it was degassed for 10 min. A condenser, thermometer, and stirring rod were then inserted into the three-necked flask, which was placed into a 70 °C constant-temperature water bath to heat and a constant-temperature reaction for 10 h under the condition of setting the stirring rate to 50 rpm. The molecularly imprinted polymer (MIP) was collected, and the template molecules and non-polymerized compounds were extracted simply from MIP microspheres by cleaning with a methanol/acetic acid mixture (9:1, *v*/*v*). Then, it was air-dried at 60 °C in an oven (FD115, Binder, Tuttlingen, Germany) for 12 h and stored in a desiccator. The non-imprinted polymer (NIP) microspheres were also prepared with the same procedure without the template molecule added to the reaction mixture.

For the encapsulation of MIP microspheres in electrospinning nanofibers, we heated 8% polyvinyl alcohol (PVA/water, *w*/*v*) in a water bath at 90 °C for 1 h to completely dissolve the PVA solution, then added 0.2% MIP to the methanol solution and ultrasonically dispersed it for 1 h to make it uniformly suspended in methanol. Then, PVA and MIP solutions of the same volume were mixed and stirred in a 60 °C water bath for 1 h to obtain a uniformly dispersed electrospinning solution. After that, the mixture was electrospun, the voltage was set to 20 kV, the speed was set to 1 mL·h^−1^, and the iron plate collector was positioned 15 cm from the tip of the syringe. After 10 h of spinning, a molecularly imprinted composite membrane (MICM) was obtained. A non-imprinted composite membrane (NICM) was also prepared under the same conditions, except that the MIP in the electrospinning solution was replaced by NIP.

#### 2.2.2. Preparation of Molecularly Imprinted Nanofiber Membranes

Molecularly imprinted nanofiber membranes (MINMs) were directly prepared by the electrospinning technique. PVA (1.2000 g), EGMRA (0.1200 g), and the template molecule chrysin (0.0375 g) were dissolved in a DMF/water solution (2:1, *v*/*v*). The mixture was continuously agitated in a closed vial for at least 2 h until no phase separation was observed. The solution was transferred to a 10 mL syringe installed with a metal needle that had an inner diameter of 0.8 mm. After that, the mixture was electrospun, the voltage was set to 20 kV, the speed was set to 1 mL·h^−1^, and the iron plate collector was placed 15 cm from the tip of the syringe. After 10 h of spinning, the fibrous nanofibers were collected on the iron plate collector. After methanol-solvent extraction, nanofibers were inspected with UV–Vis spectrometers (UV-2700, Shimadzu, Kyoto, Japan) to determine that the template molecule chrysin was no longer detectable from the washing solvent. Afterward, these nanofibers were kept in a vacuum chamber for 24 h to eliminate trace solvents, then stored in a desiccator. In comparison, the non-imprinted nanofiber membranes (NINMs) were also spun in the same way without adding chrysin.

### 2.3. Characterization

The surface morphology of MINM and NINM after the samples were sprayed with gold was observed using a scanning electron microscope (SEM, SUPRA 55 Sapphire, Carl Zeiss Jena, Jena, Germany) under low vacuum conditions. Fourier transform infrared (FT-IR, MAGNA-IR550, Thermo Fisher Scientific, Waltham, MA, USA) spectra of membranes were measured with an infrared spectrometer with a wavenumber range of 4000–400 cm^−1^. The thermogravimetric analysis (TGA) and differential thermal analysis (DTA) studies were performed using a thermal gravimetric analyzer (STA449F3, Netzsch-Gerätebau GmbH, Selb, Germany) [47]. The temperature was increased from 25 to 700 °C under a nitrogen atmosphere with a heating rate of 10 °C min^−1^.

### 2.4. Adsorption Experiments

#### 2.4.1. Adsorption Kinetics

To research the adsorption kinetics of membranes for chrysin, we dispersed 20 mg of the adsorbent samples into 20 mL of chrysin methanol solution (3.9333 mM), which was oscillated for 15, 30, 45, 60, 75, 90, 105, 120, 150, 180, 210, 240, and 300 min at room temperature. At each set time points, the concentration of chrysin in the solution was measured, and the adsorption mass of chrysin was calculated. The binding capacities of membranes were calculated according to the formula [48]:(1)$$Q_{t}=\frac{\left(C_{0}-C_{t}\right)\times V}{m}$$
where *C*_0_ (mM) is the initial chrysin concentration and *C_t_* (mM) is the concentration of chrysin solution at time *t* (min). *V* (mL) is the volume of chrysin solution, and *m* (g) is the mass of membranes.

#### 2.4.2. Adsorption Isotherm

To understand the controlling mechanisms and to quantify the maximum adsorption capacity of adsorbents, 20 mg of each of the membranes was added to 20 mL of chrysin solution with different concentrations and oscillated for 5 h, in which the initial concentrations were various (0.7867, 1.5733, 2.3600, 3.1467, and 3.9333 mM). We measured the concentration of chrysin in the solution after the adsorption was over. The equilibrium adsorption capacity was calculated using the following equation [49]:(2)$$Q_{e}=\frac{\left(C_{0}-C_{e}\right)\times V}{m}$$
where *C*_0_ (mM) is the initial chrysin concentration, and *C_e_* (mM) is the equilibrium chrysin concentration. *V* (mL) is the volume of chrysin solution, and *m* (g) is the mass of membranes.

### 2.5. Adsorption Selectivity

To investigate the selectivity of membranes to chrysin, chloramphenicol and oxytetracycline were chosen as the compared molecules. The methanol solutions (3.9333 mM) of chrysin, chloramphenicol, and oxytetracycline were prepared, respectively, and then 20 mL of the solution was added to an Erlenmeyer flask, and then 20 mg of the adsorbent sample was added for 5 h at room temperature with shaking. Measure the concentration of chrysin, chloramphenicol, or oxytetracycline in the different solutions after the adsorption was over. The calculation formula of equilibrium adsorption capacity was the same as Formula (2).

### 2.6. Adsorption Reusability

After the adsorbent sample completed the adsorption process, the saturated sample was obtained by filtration. The filtered samples were washed with methanol and dried to obtain the regenerated MINM. The regenerated MINM was reused for the next adsorption test. Under the same conditions, the adsorption-desorption cycle was repeated 6 times, and the adsorption amount was measured and calculated each time.

### 2.7. Mechanical Properties

The mechanical properties of the MINM were measured by using an electromechanical universal testing machine (JDL-10000N, Yangzhou Tianfa Testing Machinery Co., Ltd., Yangzhou, China) at room temperature. The MINM was cut into strips (20 mm × 10 mm), and its thickness was measured at different locations with a micrometer. The MINM was clamped at both ends and stretched along its length at a certain tensile rate until broken. By averaging the results of three parallel experiments, the tensile strength and elongation at rupture of the MINM were determined. The tensile strength was the load per unit area when the sample was broken on the tensile machine, expressed in *P*, which was defined as follows [50,51,52,53]:(3)$$P=\frac{F}{S}$$
where *P* (MPa) is the tensile strength of the MINM, *F* (N) is the force on the fractured section when the MINM broke, and *S* (mm^2^) is the area of the fracture surface of the MINM.

## 3. Results and Discussion

### 3.1. Optimization of Preparation Conditions of MINM

To investigate the effects of template molecule content, rosin ester content, and electrospinning voltage on the adsorption capacity of nanofiber membranes, different single-factor optimization experiments were carried out.

First, different MINMs were prepared by changing the content of template molecules (chrysin) in the spinning solution to determine the optimal content of template molecules, while the other preparation processes remained the same. As can be seen from Figure 1a, it could be found that with the increasing concentration of chrysin in the spinning solution, the MINM has an increased adsorption capacity to chrysin. When the usage content of chrysin was 0.25%, the optimum adsorption capacity of the MINM was successfully prepared, which should be attributed to the production of numerous imprinting cavities and recognition loci. There was, however, an obvious decrease in the adsorption ability of the MINM after increasing the chrysin content in the spinning solution continuously, which could be because the competitive synthesis loci of chrysin occurred when the amount of the chrysin was in excess, resulting in the decrease in the imprinting effect.

Second, to acquire the optimal concentration of rosin ester in the preparation of the MINM, MINMs with different rosin ester contents were prepared with the other conditions unchanged. As observed in Figure 1b, the MINM prepared with different contents of rosin ester showed different adsorption capacities. The adsorption capacity of the prepared MINM increased as the content of rosin ester increased. An appropriate amount of rosin ester enhanced the rigidity and mechanical properties of the MINM and maintained the spatial structure and cavity of the MINM, thereby improving the specific binding ability to the imprinted loci. When the content of rosin ester reached 10%, the adsorption capacity reached the maximum. However, superabundant rosin ester will increase the viscosity, resulting in a decrease in the pore size of the prepared MINM and membrane flux, which will make it difficult for chrysin to reach the binding cavities in the MINM, resulting in a decrease in adsorption capacity. Therefore, the MINM with a 10% addition of rosin ester had the best adsorption capacity of all the MINMs.

Third, the effect of the electrospinning voltage on the adsorption capacity during the preparation of MINMs was then studied by ranging the voltage from 15 kV to 30 kV. It can be seen in Figure 1c that MINMs prepared with different spinning voltages exhibited different adsorption capacities. As shown, when the voltage reached 20 kV, the optimum adsorption capacity of the MINM toward chrysin was achieved; however, continuing to increase the voltage reduces the adsorption capacity. This could be because as the voltage increased, the structure of the prepared nanofiber membrane became more uniform, meaning that the surface of the membrane adsorbed chrysin more easily. In contrast, when the voltage was applied over 20 kV, the adsorption capacity of the MINM toward chrysin reduced gradually. Therefore, the optimal voltage in the spinning process was 20 kV.

### 3.2. Morphology of MINM

The surface characteristics of the membranes were investigated by SEM, and the results are shown in Figure 2a,c,e. The SEM photographs illustrate that the MINM and NINM had an appreciable difference in morphology and fiber diameter. The MICM had microsphere particles embedded in the fibers. The diameter distributions of the MINM, NINM, and MICM are shown in Figure 2b,d,f; the average diameter of 80 of the randomly selected nanofibers in the MINM was about 489 nm. However, the average diameter of the nanofibers in the NINM was nearly 246 nm. The average diameter of the nanofibers in the MICM was 205 nm. The MINM provides an excellent surface area and porosity to enable the transport of the chrysin molecules through the membranes. This network structure of membranes supports the easy diffusion of chrysin molecules across the membrane surface, which provides a high potential for chrysin recognition applications. According to these results, the different recognition behaviors of the MINM toward chrysin are caused by the efficient footprints and not the morphological differences.

To provide evidence for the process of the imprinting of chrysin, the MINM, NINM, and unwashed MINM were compared, as shown in Figure 2g. The FT-IR of the washed MINM and NINM exhibited the semblable shapes, which indicated that these membranes had a similar backbone. The peak at 3274 cm^−1^ in the infrared spectrum belongs to the stretching vibration of the hydroxyl group (-OH) in the PVA, and the peak at 2945 cm^−1^ belongs to the stretching vibration of the methylene group (-CH2-) in the PVA. Infrared spectroscopy analysis was performed on the unwashed MINM and the washed MINM to study the interaction between PVA and chrysin. As shown in Figure 2g, after chrysin was added, a vibration stretch peak appeared at 1617 cm^−1^, which represents the stretch peak of the benzene ring skeleton on the chrysin, but the corresponding peak did not appear in the MINM after washing, indicating that the chrysin was successfully washed. The infrared comparison between the MINM and NINM after washing showed that there was a displacement at 1734 cm^−1^, which may be caused by the formation of hydrogen bonds between PVA and chrysin. These results indicated that the combination of chrysin and the membrane material was successful, and chrysin was removed successfully with methanol solution. The FTIR of the MICM, the peak at 3274 cm^−1^ belongs to the stretching vibration of the hydroxyl group (-OH) in the PVA, and the peak at 2945 cm^−1^ belongs to the stretching vibration of the methylene group (-CH2-) in the PVA. The peak at 1734 cm^−1^ belongs to the stretching vibration of the carbonyl group (C=O) in the molecularly imprinted polymer. Combined with SEM and FTIR of the MICM, it can be seen that the molecularly imprinted polymer was successfully incorporated into the MICM.

The thermostability of the MINM and MICM was tested by TGA analysis, as shown in Figure 2h. The MINM began to decompose at approximately 275 °C, the decomposition rate reached the maximum level when the temperature was about 361 °C, and at the end of the thermal decomposition process, the temperature reached 480 °C. However, the MICM began to decompose at approximately 255 °C, the decomposition rate reached the maximum level when the temperature was about 345 °C, and at the end of the thermal decomposition process, the temperature reached 500 °C. The reason for decomposition was mainly ascribed to the scission of the main chain and the scission of cross-linked bonds. The TGA results show that the MINM had excellent thermostability. This was ascribed to the characteristic hydrocarbon-based phenanthrene rings of the EGMRA, which raised the thermostability of the MINM.

### 3.3. Adsorption Kinetics

In order to clarify the adsorption rate control mechanism of the adsorption process, the adsorption kinetics experiment was carried out on the MINM, NINM, MICM, and NICM under the condition of the initial concentration of chrysin solution of 3.9333 mM, and the results of the experiment are shown as Figure 3a. In the four adsorbent samples, the adsorption capacity was raised with the increase in time. The adsorption capacity of the MINM for chrysin was significantly greater than that of the NINM, MICM, and NICM, while the adsorption capacity of the MICM was significantly greater than that of the NICM. In the first 90 min, the adsorption was in the rapid adsorption stage, the adsorption capacity reached about 80% of the maximum adsorption capacity, and the adsorption reached equilibrium after 120 min, the maximum adsorption capacity of the MINM was 127.5 mg·g^−1^, and the adsorbed chrysin amounts in this study were much higher than those reported previously [54]. It was because, in the early stage of adsorption, the concentration of chrysin was higher, and the molecular diffusion rate was faster. Numerous binding loci on the surface of the MINM and MICM quickly and specifically adsorbed chrysin. When the binding loci on the surface reached saturation, the adsorption rate gradually slowed down. There were no imprinting binding loci matching with chrysin in the NINM and NICM, so the adsorption rate was relatively slow, and the adsorption capacity to chrysin was low. In comparison to the MICM, the MINM exhibited higher adsorption capacity. because, in the MICM, the chrysin access to imprinted loci was much more difficult. Many MIP particles were not on the surface of the fibers but inside and so less accessible.

Moreover, pseudo-second-order and pseudo-first-order models were used for fitting kinetic curves to study adsorption mechanisms, such as physical adsorption and chemical adsorption [55].

The pseudo-first-order kinetic model was calculated by the equation:(4)$$ln\left(Q_{e}-Q_{t}\right)=lnQ_{e}-\frac{k_{1}}{2.303}t$$

The pseudo-second-order kinetic model was calculated by the equation:(5)$$\frac{t}{Q_{t}}=\frac{1}{k_{2}Q_{e}^{2}}+\frac{t}{Q_{e}}$$
where *Q_t_* (mg·g^−1^) is the adsorption quantity at time *t* (h), *Q_e_* is the amount of chrysin absorbed at equilibrium, and *k*_1_ and *k*_2_ are the equilibrium rate constant of two kinetics models.

The fitting curves of the two kinetic models are shown in Figure 3b,c. The pseudo-first-order kinetic theory predicts that adsorption loci occupancy is proportional to the unoccupied loci, whereas, in the pseudo-second-order kinetic model, the adsorption rate is determined by chemisorption between the template molecule and the adsorbent. The corresponding parameters of the two kinetic equations are determined and exhibited in Table 1.

The correlation coefficient of the MICM’s pseudo-second-order kinetic model (R^2^ = 0.9998) is larger than that of the pseudo-first-order kinetic model (R^2^ = 0.9524). It indicates that the adsorption process was more in line with the pseudo-second-order kinetic model, and the adsorption rate was affected by the binding ability of chrysin and the quantity of imprinting binding loci, which indicates that chemical interaction plays a leading role in the chrysin adsorption process. Similarly, the pseudo-second-order kinetic model’s correlation coefficient (R^2^ = 0.9984) of the MINM was higher than that of the pseudo-first-order kinetic model (R^2^ = 0.9538) of the MINM. These results indicate that physical and chemical adsorption existed in the adsorption process, but chemisorption was prevailing. These results correlate with the high surface area and porosity of the MINM, the effective molecularly imprinted cavities due to chrysin on the surfaces of the MINM, and the non-covalent interaction between the MINM and chrysin.

### 3.4. Adsorption Isotherm

In order to measure the adsorption behavior of molecularly imprinted nanofiber membranes and molecularly imprinted composite membranes, the isotherms studies were performed, as shown in Figure 4a. With an increasing chrysin concentration, equilibrium adsorption capacity increased. It was indicated that molecularly imprinted membranes’ binding capacity was better than that of non-molecularly imprinted membranes in the same situation, which implied the existence of abundant recognition loci and affinity capacity for template molecular (chrysin) on the surface of the molecularly imprinted membranes. However, under the same conditions of two kinds of molecularly imprinted membranes, the binding capacity of the MINM was significantly higher than that of the MICM. It may be because there were more recognition loci on the surface.

In addition, adsorption equilibrium data of membranes used two typical isotherm models for adsorption [48]. The Langmuir isotherm model assumes the existence of monolayer adsorption onto a surface with a limited number of binding loci, and the Freundlich isotherm model assumes the exponential distribution of adsorption loci on the multilayer adsorption. The two isotherm models were mathematically described as follows:

Langmuir isotherm:(6)$$\frac{1}{Q_{e}}=\frac{1}{Q_{m}}+\frac{1}{k_{3}Q_{m}}\times \frac{1}{C_{e}}$$

Freundlich isotherm:(7)$$lnQ_{e}=lnk_{4}+\frac{1}{n}lnC_{e}$$
where *Q_m_* (mg·g^−1^) is the greatest adsorption quantity, *Q_e_* (mg·g^−1^) is the chrysin adsorption quantity at different initial concentrations, *C_e_* (mM) is the equilibrium concentration, *k* is the constant, and 1/*n* is the heterogeneity factor indicating adsorption intensity.

Two kinds of adsorption isotherm for the MINM and MICM are shown in Figure 4b,c. Additionally, various kinds of isotherm parameters are shown in Table 2. It is observed that the MINM and MICM had correlation coefficients of 0.9976 and 0.9999, respectively, for the Langmuir adsorption isotherm, while MINM and MICM had correlation coefficients of 0.9921 and 0.9926, respectively, concerning the Freundlich adsorption isotherm. The results show that, in the studied concentration range, the adsorption of chrysin matches the Langmuir model better than the Freundlich model. Meanwhile, both MINM and MICM had a 1/n value in the range of 0.5–1, illustrating that the two kinds of membranes are excellent adsorption materials for chrysin.

### 3.5. Adsorption Selectivity

The adsorption selectivity is an essential characteristic for the application of MIMs. Thus, to examine the selectivity of MIMs to chrysin, chloramphenicol and oxytetracycline were chosen as comparative substrates in the selective adsorption test. These two molecules have similar structures and functional groups to chrysin. The selectivity research was carried out on chrysin and its comparative substrates at the concentration of 3.9333 mM. As shown in Figure 5a, the MINM had adsorption capacity for all three substances, but the adsorption capacity of the MINM for chrysin was significantly higher than for other molecules. Although the adsorption capacity of the MICM for chrysin was also better than the compared molecules, the adsorption capacity was lower than that of the MINM. The NINM and NICM had poor adsorption capacity for the three substances, and there was not much difference. All of the above results indicate that chrysin was able to selectively adsorb the MINM and MICM due to the imprinting cavities formed during the preparation in the presence of chrysin, which led to the formation of affinity binding loci along with access in the MINM and MICM. The shape, size, and functional group of these recognition loci form complementary structures to chrysin. It is profitable that the MINM and MICM have an affinity for binding chrysin. However, the NINM and NICM do not have the related loci and the recognition capability coming from the imprinting effect. It is, therefore, possible to conclude that the imprinting loci on the surface of the MINM and MICM have excellent selectivity for chrysin, and the recognition ability is provided by the imprinting loci, and compared with two printing membranes, the MINM has a stronger selective recognition ability for chrysin. The results of this study not only have better selectivity but also higher chrysin adsorption capacity compared to previous reports [56,57].

### 3.6. Adsorption Reusability

Besides selectivity, stability and reusability are also important indexes to evaluate the performance of the MINM. To evaluate the capacity of the MINM to be regenerated and reused, the adsorption performance after repeated cycles was investigated. After each binding experiment, the MINM was washed with methanol/acetic acid solution (9:1, *v*/*v*) to remove the adsorbed molecules. Hence, we proceeded to the next adsorption cycle. The above processes were repeated until the sorption-desorption was accomplished in six cycles. The results are illustrated in Figure 5b. The results suggest that the MINM exhibited excellent adsorption capability in all six cycles. After being recycled and reused, the MINM only lost 12.08% of adsorption capacity. This decrease may be attributed to the reduction in active binding loci following regeneration and inadequate desorption of the adsorbed chrysin molecules. It indicates that the MINM could be used repeatedly due to its stability and reusability.

### 3.7. Mechanical Properties of MINM

A special focus was placed on the mechanical properties of the membranes, since these were related to stress levels experienced during operation. The tensile strength and elongation at break of the MINM were tested, as shown in Figure 5c. The maximum tensile strength of the MINM was 6.5 MPa, and the breaking elongation of the MINM was 130%, while the maximum tensile strength of PVA nanofiber was only 1.58 MPa [58], the MINM showing excellent mechanical properties. This result indicates that this MINM enables the higher sustainability of the membrane for use in some special operating environments.

## 4. Conclusions

In this study, we fabricated a molecularly imprinted nanofiber membrane of chrysin using an electrospinning method for the selective adsorption of chrysin molecules. Compared with the molecularly imprinted composite membrane (MICM), the prepared MINM has a larger specific surface area, more binding sites complementary to the template are generated near the surface of the fiber, showing high adsorption capacity and significant selectivity, and the performance is generally better than that of the MICM. The adsorption kinetics suggested that the adsorption process of the MINM was more consistent with the pseudo-second-order kinetic model, illustrating that the adsorption process was controlled by chemisorption. The adsorption isotherms illustrated that the adsorption process of chrysin is in accordance with the Langmuir model rather than the Freundlich model. In addition, the MINM exhibited good thermal stability and excellent reusability. The conspicuous adsorption behavior coupled with the effortless preparation made the MINM a potential candidate for the adsorption of chrysin. In summary, we consider that this method provides a low-cost, effective way for efficiently separating and enriching the chrysin.

## Figures and Tables

**Figure 1 polymers-14-02398-f001:**
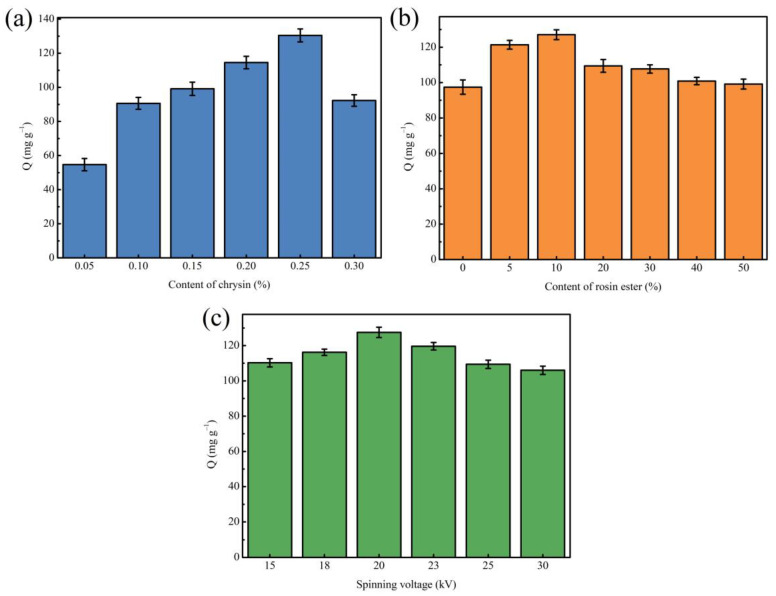
(**a**) Effect of different content of chrysin on the adsorption capacity of MINM; (**b**) effect of different rosin ester content on the adsorption capacity of MINM; (**c**) effect of different spinning voltage on the adsorption capacity of MINM.

**Figure 2 polymers-14-02398-f002:**
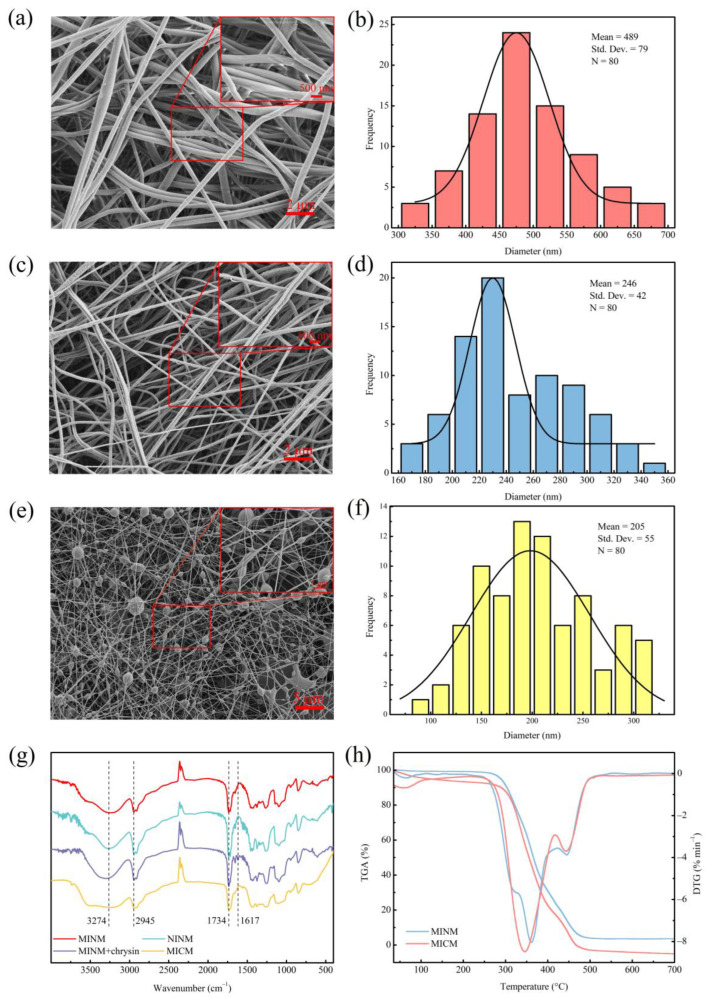
(**a**) SEM micrographs of MINM; (**b**) diameter distribution of MINM; (**c**) SEM micrographs of NINM; (**d**) diameter distribution of NINM; (**e**) SEM micrographs of MICM; (**f**) diameter distribution of MICM; (**g**) FT-IR spectra of the MINM, NINM, unwashed MINM, and MICM; (**h**) TGA and DTG curves of MINM and MICM.

**Figure 3 polymers-14-02398-f003:**
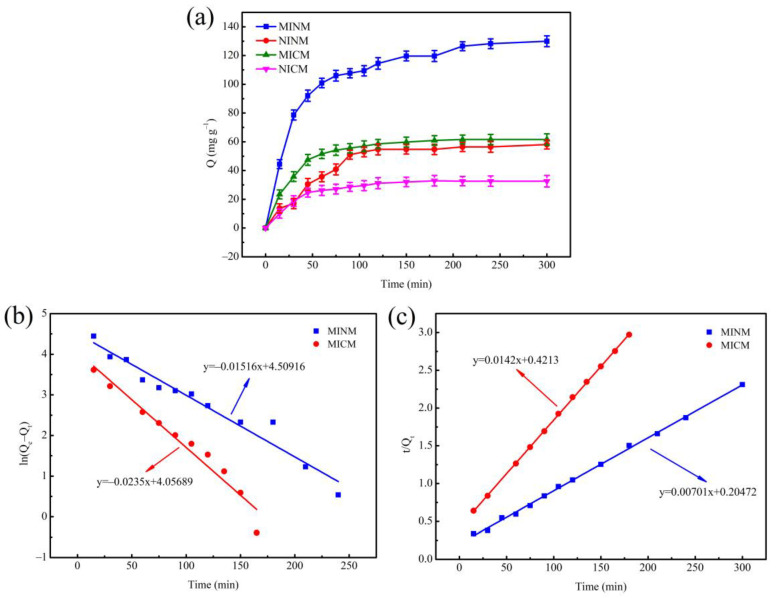
(**a**) The adsorption kinetics of MINM, NINM, MICM, and NICM; (**b**) the pseudo-first-order kinetic model of MINM and MICM; (**c**) the pseudo-second-order kinetic model of MINM and MICM.

**Figure 4 polymers-14-02398-f004:**
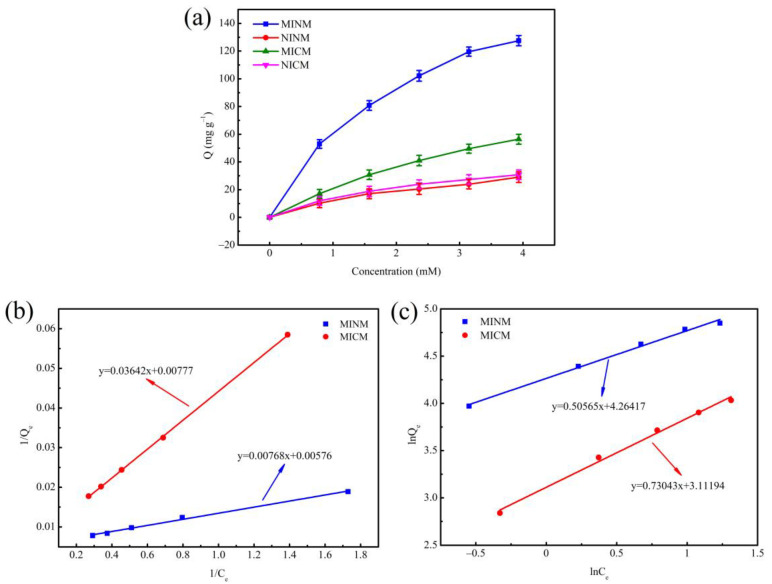
(**a**) The adsorption isotherms of MINM, NINM, MICM, and NICM; (**b**) the Langmuir isotherm model of MINM and MICM; (**c**) the Freundlich isotherm model of MINM and MICM.

**Figure 5 polymers-14-02398-f005:**
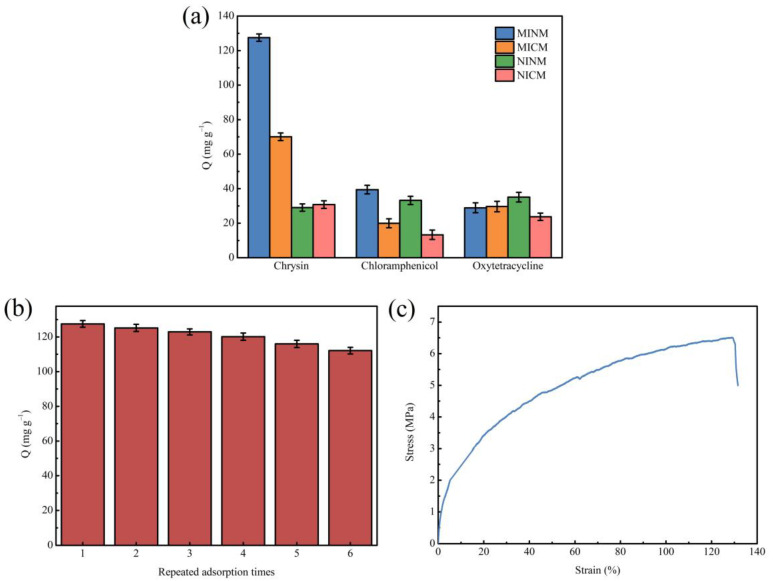
(**a**)The selective adsorption capacity of MINM, MICM, NINM, and NICM; (**b**) regeneration rebinding performance of MINM; (**c**) stress–strain curve of MINM.

**Table 1 polymers-14-02398-t001:** Kinetic data of pseudo-first-order kinetic model and pseudo-second-order kinetic model.

Samples	Pseudo-First-Order Kinetic		Pseudo-Second-Order Kinetic	
	k_1_ (min^−1^)	R^2^	k_2_ (g·mg^−1^ min^−1^)	R^2^
MINM	0.0349	0.9538	0.24 × 10^−3^	0.9984
MICM	0.0541	0.9524	0.48 × 10^−3^	0.9998

**Table 2 polymers-14-02398-t002:** Parameters of Langmuir adsorption model and Freundlich adsorption model.

Samples	Langmuir Isotherm			Freundlich Isotherm		
	k_3_ (mM^−1^)	R^2^	Q_m_ (mg·g^−1^)	k_4_ (mM^−1^)	R^2^	1/n
MINM	0.7500	0.9976	173.611	71.1059	0.9921	0.5057
MICM	0.2133	0.9999	128.700	22.4646	0.9926	0.7304

## Data Availability

Not applicable.

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
