# Peer review of "Preparation and Recognition Properties of Molecularly Imprinted Nanofiber Membrane of Chrysin"

_polymers, 2022, doi:10.3390/polym14122398_

Round 1
Reviewer 1 Report
The paper entitled 'Preparation and Recognition Properties of Molecularly Imprinted Nanofiber membrane of Chrysin' studies the possibility of using rosin- based molecularly imprinted membranes for the adsorption of chrysin. The article is well structured. However, there are some improtant points needed to be fixed before acceptance.
-Abstract: first two sentence are the same and misunderstood.
- The authors mentioned several times large specific surface area. On the basis of which you have concluded this statement? Could you please specify which one was?
- Results and discussion need to be compared with current literature
- the authors didn't give the SEM micrographs, FTIR and TGA analysis for MICM
Author Response
Comments:
The paper entitled 'Preparation and Recognition Properties of Molecularly Imprinted Nanofiber membrane of Chrysin' studies the possibility of using rosin- based molecularly imprinted membranes for the adsorption of chrysin. The article is well structured. However, there are some improtant points needed to be fixed before acceptance.
Point 1: Abstract: first two sentence are the same and misunderstood.
Response 1: Sorry for our carelessness. The Abstract part has been modified accordingly, and the first two sentence of the abstract have been modified into one sentence.
Point 2: The authors mentioned several times large specific surface area. On the basis of which you have concluded this statement? Could you please specify which one was?
Response 2: I'm very sorry for the difficulty in reading caused by language expression. Generally speaking, electrospinning is a technique for producing ultra-fine nanofibers, so electrospun fiber membrane has the characteristics of a high specific surface area and high porosity. Moreover, it can be seen from the SEM that the inner space of MINM is intricate, and the surface of the fiber has plenty of grooves and wrinkles, which increases the specific surface area.
Point 3: Results and discussion need to be compared with current literature.
Response 3: Many thanks for this suggestion. We added a comparison with current literature in the results and discussion, and cited the following articles in the revised manuscript:
Iben Nasser, I.; Algieri, C.; Garofalo, A.; Drioli, E.; Ahmed, C.; Donato, L. Hybrid imprinted membranes for selective recognition of quercetin. Separation and Purification Technology 2016, 163, 331-340.
Liang, C.; Zhang, Z.; Zhang, H.; Ye, L.; He, J.; Ou, J.; Wu, Q. Ordered macroporous molecularly imprinted polymers prepared by a surface imprinting method and their applications to the direct extraction of flavonoids from Gingko leaves. Food Chemistry 2020, 309, 125680.
Li, X.; Dai, Y.; Row, K.H. Preparation of two-dimensional magnetic molecularly imprinted polymers based on boron nitride and a deep eutectic solvent for the selective recognition of flavonoids. Analyst 2019, 144, 1777-1788.
Ullah, S.; Hashmi, M.; Hussain, N.; Ullah, A.; Sarwar, M.N.; Saito, Y.; Kim, S.H.; Kim, I.S. Stabilized nanofibers of polyvinyl alcohol (PVA) crosslinked by unique method for efficient removal of heavy metal ions. Journal of Water Process Engineering 2020, 33, 101111.
Point 4: the authors didn't give the SEM micrographs, FTIR and TGA analysis for MICM.
Response 4: Sorry for our carelessness. We have added the SEM micrographs, diameter distribution, FTIR and TGA analysis for MICM in the revised manuscript.

Reviewer 2 Report
Review of a manuscript entitled “Preparation and Recognition Properties of Molecularly Imprinted Nanofiber Membrane of Chrysin”
The authors of this study describe electrospinning-based formation of molecularly imprinted nanofiber membrane imprinted with chrysin. Chrysin is a flavonoid that exhibits anti-oxidation, anti-viral, immunomodulatory and anti-inflammatory effects. In this study are compared two methods of preparation of these membranes based onelectrospinning.
The idea of the manuscript is interesting but the description is poor and all parts of the manuscript must be significantly advanced and improved.
The abstract should be improved.
Each method described in the introduction must be supported by appropriate references.
English grammar rules should be followed.
Keywords are not representative and should be reviewed.
Experimental part: The methodology should be significantly improved.
The purpose of the manuscript is not clear.
Abbreviations must be clearly described in the manuscript.
Rosin-based molecularly imprinted membranes are mentioned in the abstract, but according to the experimental description it was used a monomer ethylene glycol maleic rosinate acrylate. So, to call the membrane as rosin based is not enough accurate.
Recent Reviews (Advances in molecularly imprinted polymers based affinity sensors (Review). Polymers 2021, 13, 974. // https://doi.org/10.1016/j.snb.2020.128973 ) on molecularly imprinted polymers and determination of chrysin by other methods (https://doi.org/10.1111/j.1541-4337.2010.00130.x ; https://doi.org/10.1016/j.fct.2008.12.007 ; https://doi.org/10.3390/antiox11040645; http://dx.doi.org/10.1016/j.chroma.2004.10.068 ) could be overviewed and discussed.
The journal template should be applied for the manuscript.
CAS numbers should be added to the descriptions of the chemicals.
It is more usual to write the concentrations of monomers in mM.
The list of references should be extended.
Check for correct name for wood butterflies. The name of species must be written in English and Latin too.
Author Response
Comments:
Review of a manuscript entitled “Preparation and Recognition Properties of Molecularly Imprinted Nanofiber Membrane of Chrysin”
The authors of this study describe electrospinning-based formation of molecularly imprinted nanofiber membrane imprinted with chrysin. Chrysin is a flavonoid that exhibits anti-oxidation, anti-viral, immunomodulatory and anti-inflammatory effects. In this study are compared two methods of preparation of these membranes based on electrospinning.
The idea of the manuscript is interesting but the description is poor and all parts of the manuscript must be significantly advanced and improved.
Point 1: The abstract should be improved.
Response 1: Thank you for your suggestion. The abstract part has been modified accordingly in the revised manuscript.
Point 2: Each method described in the introduction must be supported by appropriate references.
Response 2: Sorry for our carelessness. We have added appropriate references to each method described in the introduction in the revised manuscript.
Point 3: English grammar rules should be followed.
Response 3: Sorry for our carelessness. We found some grammatical errors through re-examination and made corresponding changes, then used the editing service at https://www.mdpi.com/authors/english.
Point 4: Keywords are not representative and should be reviewed.
Response 4: Thank you for your suggestion. We have made corresponding modifications for the keywords.
Point 5: Experimental part: The methodology should be significantly improved.
Response 5: Thank you for your suggestion. We have revised accordingly in the revised manuscript.
Point 6: The purpose of the manuscript is not clear.
Response 6: Sorry for our carelessness. We have revised the description regarding to this issue in the revised manuscript.
Point 7: Abbreviations must be clearly described in the manuscript.
Response 7: Sorry for our carelessness. We describe each abbreviation clearly when it first appears.
Point 8: Rosin-based molecularly imprinted membranes are mentioned in the abstract, but according to the experimental description it was used a monomer ethylene glycol maleic rosinate acrylate. So, to call the membrane as rosin based is not enough accurate..
Response 8: Thank you for your suggestion. We have made corresponding modifications for its name.
Point 9: Recent Reviews (Advances in molecularly imprinted polymers based affinity sensors (Review). Polymers 2021, 13, 974. // https://doi.org/10.1016/j.snb.2020.128973) on molecularly imprinted polymers and determination of chrysin by other methods (https://doi.org/10.1111/j.1541-4337.2010.00130.x; https://doi.org/10.1016/j.fct.2008.12.007; https://doi.org/10.3390/antiox11040645; http://dx.doi.org/10.1016/j.chroma.2004.10.068) could be overviewed and discussed.
Response 9: Thank you for your suggestion. We refer to the following articles:
Ramanavicius, S.; Jagminas, A.; Ramanavicius, A. Advances in Molecularly Imprinted Polymers Based Affinity Sensors (Review). Polymers (Basel) 2021, 13.
Saric, A.; Balog, T.; Sobocanec, S.; et al. Antioxidant effects of flavonoid from Croatian Cystus incanus L. rich bee pollen. Food and Chemical Toxicology 2009, 47, 547-554.
Kaskoniene, V.; Venskutonis, P.R. Floral Markers in Honey of Various Botanical and Geographic Origins: A Review. Comprehensive Reviews in Food Science and Food Safety 2010, 9, 620-634.
Molnar-Perl, I.; Fuzfai, Z. Chromatographic, capillary electrophoretic and capillary electrochromatographic techniques in the analysis of flavonoids. Journal of Chromatography A 2005, 1073, 201-227
Lowdon, J.W.; Dilien, H.; Singla, P.; et al. MIPs for commercial application in low-cost sensors and assays - An overview of the current status quo. Sensors and Actuators B-Chemical 2020, 325, 128973.
Adaskeviciute, V.; Kaskoniene, V.; Barcauskaite, K.; et al. The Impact of Fermentation on Bee Pollen Polyphenolic Compounds Composition. Antioxidants (Basel) 2022, 11, 645.
Point 10: The journal template should be applied for the manuscript.
Response 10: Thank you for your suggestion. We have revised the manuscript base on the journal template.
Point 11: CAS numbers should be added to the descriptions of the chemicals.
Response 11: Yes, we have added the CAS numbers when descriptions of each chemical.
Point 12: It is more usual to write the concentrations of monomers in mM.
Response 12: Thank you for your suggestion. We have used mM as the concentrations of monomers and changed the graph accordingly.
Point 13: The list of references should be extended.
Response 13: Thank you for your suggestion. We have revised accordingly. We extended with 13 references.
Point 14: Check for correct name for wood butterflies. The name of species must be written in English and Latin too.
Response 14: Sorry for our carelessness. The correct name should be “oroxylum indicum”, and we have revised accordingly in the manuscript.

Reviewer 3 Report
The English needs correction. For instance this part:
To research the adsorption kinetics of membranes for chrysin. Disperse 20 mg of the adsorbent samples into 20 mL of methanol solution…..
How do you measure the concentration of Chrysin for adsorption calculation?
I do not understand how do you measure the selectivity by your protocol below:
To investigate the selectivity of membranes to chrysin, chloramphenicol and oxytetracycline were chosen as the compared molecules. The methanol solutions (1 mg mL-1) of chrysin, chloramphenicol and oxytetracycline were prepared respectively, and then 20 mL of the solution was put into the Erlenmeyer flask, and then 20 mg of the adsorbent samples were added for 5 hours at room temperature with shaking.. Something is missing!
Also the English is not good in the part bellow:
After the adsorbent samples completed the adsorption process, it was filtered to obtain a saturated sample, then washed with methanol solution, dried and weighed to obtain MINM. The regenerated MINM was reused for the next adsorption test. Under the same conditions, the adsorption-desorption cycle was repeated 6 times, and the adsorption amount was measured and calculated each time.
There are missing too, some details, as for instance how the drying proceeds.
I do not understand the phrase:
To research the optimal synthesis factors of MINM, a detailed investigation and evaluation of several key synthesis processes such as the content of template molecule, amount of rosin ester, and the voltage during spinning.
Also this phrase:
When the content of rosin ester reached 10%, the adsorption capacity reached the maximum because an increase in the content of rosin ester would enhance the rigidity and mechanical property and keep the MINM spatial structure and cavity to improve specific binding to imprinted receptor loci.
I disagree with this assumption:
In comparison to MICM, MINM exhibited higher adsorption capacity because it had a larger specific surface area and more hydrogen bonding loci.
In my opinion the difference comes from the fact that in MICM the access to imprinted site is much more difficult, because many PMAA particles are not on the surface of the fibbers, but inside and so less accessible.
I disagree also with this part:
However, the pseudo-second-order kinetic model correlation coefficient (R2=0.9984) of the MINM was higher than the pseudo-first-order kinetic model (R2=0.9538) of the MINM. These results indicated that the adsorption process of MINM conformed to both two kinetic models well, illustrating that the adsorption process was controlled by both physical adsorption and chemisorption rather than a single adsorption mode.
In fact you have the same situation for MINM as for MICM! The chemosorption is prevailing!
Give the article to someone with a very good knowledge of English to make correction, such as an English teacher
Author Response
Point 1: The English needs correction. For instance this part:
To research the adsorption kinetics of membranes for chrysin. Disperse 20 mg of the adsorbent samples into 20 mL of methanol solution…..
Response 1: I'm very sorry for the difficulty in reading caused by language expression. We used the editing service at https://www.mdpi.com/authors/english, and made corresponding changes in the revised manuscript.
Point 2: How do you measure the concentration of Chrysin for adsorption calculation?
Response 2: We use a UV spectrophotometer to measure the absorbance of different concentrations of chrysin solution, there is a linear correlation between the absorbance and the concentration of the solution, and the standard curve of the chrysin solution can be obtained. Then by measuring the absorbance of the unknown solution and substituting it into the standard curve, the concentration of the solution can be obtained.
Point 3: I do not understand how do you measure the selectivity by your protocol below:
To investigate the selectivity of membranes to chrysin, chloramphenicol and oxytetracycline were chosen as the compared molecules. The methanol solutions (1 mg mL-1) of chrysin, chloramphenicol and oxytetracycline were prepared respectively, and then 20 mL of the solution was put into the Erlenmeyer flask, and then 20 mg of the adsorbent samples were added for 5 hours at room temperature with shaking.. Something is missing!
Response 3: Sorry for our carelessness. We have refined the experimental method for adsorption selectivit, and made corresponding changes in the revised manuscript.
Point 4: Also the English is not good in the part bellow:
After the adsorbent samples completed the adsorption process, it was filtered to obtain a saturated sample, then washed with methanol solution, dried and weighed to obtain MINM. The regenerated MINM was reused for the next adsorption test. Under the same conditions, the adsorption-desorption cycle was repeated 6 times, and the adsorption amount was measured and calculated each time.
Response 4: Sorry for our carelessness. We have revised accordingly. We have improved the experimental method for adsorption reusability, and used the editing service at https://www.mdpi.com/authors/english to revise english grammar of the manuscript.
Point 5: There are missing too, some details, as for instance how the drying proceeds.
Response 5: Sorry for our carelessness. The samples were air-dried in an oven at 60 °C and then stored in a desiccator, and we made corresponding changes in the revised manuscript.
Point 6: I do not understand the phrase:
To research the optimal synthesis factors of MINM, a detailed investigation and evaluation of several key synthesis processes such as the content of template molecule, amount of rosin ester, and the voltage during spinning.
Response 6: I'm very sorry for the difficulty in reading caused by language expression. We added the correlation analysis in the revised manuscript.
Point 7: Also this phrase:
When the content of rosin ester reached 10%, the adsorption capacity reached the maximum because an increase in the content of rosin ester would enhance the rigidity and mechanical property and keep the MINM spatial structure and cavity to improve specific binding to imprinted receptor loci.
Response 7: The Correlation analysis of this paragraph was added in the revised manuscript.
Point 8: I disagree with this assumption:
In comparison to MICM, MINM exhibited higher adsorption capacity because it had a larger specific surface area and more hydrogen bonding loci.
In my opinion the difference comes from the fact that in MICM the access to imprinted site is much more difficult, because many PMAA particles are not on the surface of the fibbers, but inside and so less accessible.
Response 8: Yes, we revised the manuscript based on your suggestion.
Point 9: I disagree also with this part:
However, the pseudo-second-order kinetic model correlation coefficient (R2=0.9984) of the MINM was higher than the pseudo-first-order kinetic model (R2=0.9538) of the MINM. These results indicated that the adsorption process of MINM conformed to both two kinetic models well, illustrating that the adsorption process was controlled by both physical adsorption and chemisorption rather than a single adsorption mode.
In fact you have the same situation for MINM as for MICM! The chemosorption is prevailing!
Response 9: Many thanks for this suggestion. We have revised the description regarding this issue in the revised manuscript. Physical and chemical adsorption existed in the adsorption process, but chemisorption was prevailing.
Point 10: Give the article to someone with a very good knowledge of English to make correction, such as an English teacher.
Response 10: Yes, we used the editing service at https://www.mdpi.com/authors/english, and made corresponding changes in the revised manuscript.

Round 2
Reviewer 1 Report
The manuscript still need English corrections. Sentences are still confused.
Statements where large specific area is mentioned must be correlated with current literature if there is no quantitative analysis.
Author Response
Response to Reviewer 1 Comments
Point 1: The manuscript still need English corrections. Sentences are still confused.
Response 1: I'm very sorry for the difficulty in reading caused by language expression. We used the editing service at https://www.mdpi.com/authors/english, and made corresponding changes in the revised manuscript.
Point 2: Statements where large specific area is mentioned must be correlated with current literature if there is no quantitative analysis.
Response 2: Many thanks for this suggestion. According to the description in the literature, electrospinning has the advantages of a large specific surface area, and we have added relevant references when mentioning the statement of large specific surface area:
Xue, J.; Wu, T.; Dai, Y.; Xia, Y. Electrospinning and Electrospun Nanofibers: Methods, Materials, and Applications. Chemical Reviews 2019, 119, 5298-5415, doi:10.1021/acs.chemrev.8b00593.
Zhao, K.; Kang, S.X.; Yang, Y.Y.; Yu, D.G. Electrospun Functional Nanofiber Membrane for Antibiotic Removal in Water: Review. Polymers (Basel) 2021, 13, 226, doi:10.3390/polym13020226.
Agrawal, S.; Ranjan, R.; Lal, B.; Rahman, A.; Singh, S.P.; Selvaratnam, T.; Nawaz, T. Synthesis and Water Treatment Applications of Nanofibers by Electrospinning. Processes 2021, 9, 1779, doi:10.3390/pr9101779.

Reviewer 2 Report
The authors made a significant improvement in the article manuscript.
But there are some minor corrections that should be made.
Comment for the authors:
1. In the abstract, you wrote, “the existing separation and extraction methods have certain defects”. I don’t agree that the word “defects” is the most suitable. I could agree if you were choosing the word “shortcomings”, “disadvantages”, “limitations”, or “drawbacks”. But even any of them have different meanings in the text. You should think and change it.
2. I don’t understand the meaning of “with safety, environmental protection, high added value” (the third line of the abstract)
3. I don’t agree with your sentence “Pharmacologically active natural products have gained unprecedented popularity in recent years”. You should specify for what exactly they gained the popularity: in analysis by electrochemical methods, development of new analysis methods by HPLC, UPLC, electrochromatography, etc. Please make it more informative and scientific.
4. oroxylum indicum: This is a scientific journal and you should use a proper way to write a botanical name (Latin name). Names should always be italicized or underlined. The first letter of the genus name is capitalized but the specific epithet is not, e.g., Lavandula angustifolia. If the meaning is clear, the generic name can be abbreviated, e.g., L. angustifolia. Please check the International Code of Nomenclature (ICN). Here is an example of how it looks in the article: Kinetics and modeling for extraction of chrysin from Oroxylum indicum seeds. Food Sci Biotechnol 24, 2045–2050 (2015). https://doi.org/10.1007/s10068-015-0272-z.
5. Please change the word anti-oxidation to anti-oxidative in the sentence: “Chrysin exhibits anti-oxidation [6], anti-viral, immunomodulatory and anti-inflammatory effects [7-9].”
6. The sentence is incorrect: “The HPLC method has the disadvantages in the extraction process of a low extraction rate and long duration.” HPLC is an analytical method and it is not proposed for extraction. Extraction is performed by other methods and then the sample is analyzed by HPLC.
7. The next sentence “other methods, such as column chromatography and other traditional methods, have a low separation effect on the structural analogs of albumin” should be changed to a more suitable example, because albumin is a high molecular weight compound and you shouldn't use it as an example for illustration in case of chrysin.
8. Here are several articles that are more suitable instead of HPLC: Optimization of Ultrasound-assisted Extraction of Quercetin, Luteolin, Apigenin, Pinocembrin and Chrysin from Flos populi by Plackett-Burman Design Combined with Taguchi Method Chiang Mai J. Sci. (2018) 45(1) (2018) 427-439 //// Kinetics and modeling for extraction of chrysin from Oroxylum indicum seeds. Food Sci Biotechnol 24 (2015) 2045–2050. https://doi.org/10.1007/s10068-015-0272-z. /////. Optimization of microwave-assisted extraction, antioxidant capacity, and characterization of total flavonoids from the leaves of Alpinia oxyphylla Miq., Preparative Biochemistry & Biotechnology, 50:1 (2020) 82-90, DOI: https://doi.org/10.1080/10826068.2019.1663535.
9. Please specify the solvent for 8% PVA in chapter 2.2.1.
10. In chapter 3.1. you write: “However, superabundant rosin ester will form a highly cross-linked structure that will make it difficult for chrysin to reach the binding cavities in MINM, resulting in a poor adsorption capacity”. The crosslinking is regulated by EGDMA, not by EGMRA. So, you should write what was the total monomer concentration and what was the crosslinker concentration in the mixture.
11. In the same chapter 3.1. your write “which might be because the high-voltage electric field affected the distribution of the π electron cloud or weakened the strength of the hydrogen bond”, but this sentence is not supported with appropriate references and experimental results. Therefore, it would be better to remove it.
12. Could you add the DOI numbers to the references?
Author Response
Response to Reviewer 2 Comments
Comments:
The authors made a significant improvement in the article manuscript.
But there are some minor corrections that should be made.
Point 1: In the abstract, you wrote, “the existing separation and extraction methods have certain defects”. I don’t agree that the word “defects” is the most suitable. I could agree if you were choosing the word “shortcomings”, “disadvantages”, “limitations”, or “drawbacks”. But even any of them have different meanings in the text. You should think and change it.
Response 1: Thank you for your suggestion. After thinking about it, we feel that “drawbacks” is more suitable, and we have revised accordingly in the revised manuscript.
Point 2: I don’t understand the meaning of “with safety, environmental protection, high added value” (the third line of the abstract)
Response 2: I'm very sorry for the difficulty in reading caused by language expression. The abstract part has been modified accordingly in the revised manuscript.
Point 3: I don’t agree with your sentence “Pharmacologically active natural products have gained unprecedented popularity in recent years”. You should specify for what exactly they gained the popularity: in analysis by electrochemical methods, development of new analysis methods by HPLC, UPLC, electrochromatography, etc. Please make it more informative and scientific.
Response 3: Many thanks for this suggestion. We have listed specific analysis methods such as high performance liquid chromatography, ultra-high performance liquid chromatography, electrochromatography, etc., and extraction processes such as membrane separation, semi-bionic extraction, high-speed countercurrent chromatography, etc., and we have revised the description regarding this issue in the revised manuscript.
Point 4: oroxylum indicum: This is a scientific journal and you should use a proper way to write a botanical name (Latin name). Names should always be italicized or underlined. The first letter of the genus name is capitalized but the specific epithet is not, e.g., Lavandula angustifolia. If the meaning is clear, the generic name can be abbreviated, e.g., L. angustifolia. Please check the International Code of Nomenclature (ICN). Here is an example of how it looks in the article: Kinetics and modeling for extraction of chrysin from Oroxylum indicum seeds. Food Sci Biotechnol 24, 2045–2050 (2015). https://doi.org/10.1007/s10068-015-0272-z.
Response 4: Sorry for our carelessness. The correct name should be “Oroxylum indicum,” we have revised accordingly.
Point 5: Please change the word anti-oxidation to anti-oxidative in the sentence: “Chrysin exhibits anti-oxidation [6], anti-viral, immunomodulatory and anti-inflammatory effects [7-9].”
Response 5: Yes, we revised the manuscript based on your suggestion, we have changed the word anti-oxidation to anti-oxidative.
Point 6: The sentence is incorrect: “The HPLC method has the disadvantages in the extraction process of a low extraction rate and long duration.” HPLC is an analytical method and it is not proposed for extraction. Extraction is performed by other methods and then the sample is analyzed by HPLC.
Response 6: Sorry for our carelessness. We have removed the incorrect sentence.
Point 7: The next sentence “other methods, such as column chromatography and other traditional methods, have a low separation effect on the structural analogs of albumin” should be changed to a more suitable example, because albumin is a high molecular weight compound and you shouldn't use it as an example for illustration in case of chrysin.
Response 7: Sorry for our carelessness. We have replaced albumin with chrysin, and we have revised the description regarding to this issue in the revised manuscript.
Point 8: Here are several articles that are more suitable instead of HPLC: Optimization of Ultrasound-assisted Extraction of Quercetin, Luteolin, Apigenin, Pinocembrin and Chrysin from Flos populi by Plackett-Burman Design Combined with Taguchi Method Chiang Mai J. Sci. (2018) 45(1) (2018) 427-439 //// Kinetics and modeling for extraction of chrysin from Oroxylum indicum seeds. Food Sci Biotechnol 24 (2015) 2045–2050. https://doi.org/10.1007/s10068-015-0272-z. /////. Optimization of microwave-assisted extraction, antioxidant capacity, and characterization of total flavonoids from the leaves of Alpinia oxyphylla Miq., Preparative Biochemistry & Biotechnology, 50:1 (2020) 82-90, DOI: https://doi.org/10.1080/10826068.2019.1663535.
Response 8: Thank you for your suggestion. We have introduced several extraction methods, such as ultrasonic-assisted extraction, ethanol-water mixed solid-liquid extraction, microwave extraction, etc., and we refer to the following articles:
Wang, B; Goldsmith, C.D; Zhao, J; Zhao, S; Sheng, Z; Yu, W. Optimization of ultrasound-assisted extraction of quercetin, luteolin, apigenin, pinocembrin and chrysin from flos populi by plackett-burman design combined with taguchi method. Chiang Mai Journal of Science 2018, 45, 427-439.
Zhou, L.; Jing, T.; Zhang, P.; Zhang, L.; Cai, S.; Liu, T.; Fan, H.; Yang, G.; Lin, R.; Zhang, J. Kinetics and modeling for extraction of chrysin from Oroxylum indicum seeds. Food Science and Biotechnology 2015, 24, 2045-2050, doi:10.1007/s10068-015-0272-z.
Niu, Q.; Gao, Y.; Liu, P. Optimization of microwave-assisted extraction, antioxidant capacity, and characterization of total flavonoids from the leaves of Alpinia oxyphylla Miq. Preparative Biochemistry Biotechnology 2020, 50, 82-90, doi:10.1080/10826068.2019.1663535.
Point 9: Please specify the solvent for 8% PVA in chapter 2.2.1.
Response 9: Sorry for our carelessness. The solvent for 8% PVA is water (PVA/water, w/v), we have revised accordingly in the revised manuscript.
Point 10: In chapter 3.1. you write: “However, superabundant rosin ester will form a highly cross-linked structure that will make it difficult for chrysin to reach the binding cavities in MINM, resulting in a poor adsorption capacity”. The crosslinking is regulated by EGDMA, not by EGMRA. So, you should write what was the total monomer concentration and what was the crosslinker concentration in the mixture.
Response 10: Many thanks for this suggestion. We have removed the wrong description and revised the description regarding this issue in the revised manuscript. We consider that superabundant rosin ester will increase the viscosity of the spinning solution, resulting in a decrease in the pore size of the prepared MINM and membrane flux, which will make it difficult for chrysin to reach the binding cavities in MINM, resulting in a decrease adsorption capacity.
Point 11: In the same chapter 3.1. your write “which might be because the high-voltage electric field affected the distribution of the π electron cloud or weakened the strength of the hydrogen bond”, but this sentence is not supported with appropriate references and experimental results. Therefore, it would be better to remove it.
Response 11: Many thanks for this suggestion. We have removed the sentence and revised the description regarding this issue in the revised manuscript.
Point 12: Could you add the DOI numbers to the references?
Response 12: Yes, we have increased the DOI numbers to the references based on your suggestion.
Reviewer 3 Report
The article is much improved. There are still some small orthography errors, which could be solved by the technical editor team.
Author Response
Response to Reviewer 3 Comments
Comments: The article is much improved.
Point 1: There are still some small orthography errors, which could be solved by the technical editor team.
Response 1: Sorry for our carelessness. We found some orthography errors through re-examination and made corresponding changes.
Round 3
Reviewer 1 Report
The editing service isn't enough for correcting English grammar. The authors should find some English reviewer and correct the sentences in manuscript.